# LRRK2 at Striatal Synapses: Cell-Type Specificity and Mechanistic Insights

**DOI:** 10.3390/cells11010169

**Published:** 2022-01-05

**Authors:** Patrick D. Skelton, Valerie Tokars, Loukia Parisiadou

**Affiliations:** Department of Pharmacology, Feinberg School of Medicine, Northwestern University, Chicago, IL 60611, USA; patrick.skelton@northwestern.edu (P.D.S.); v-tokars@northwestern.edu (V.T.)

**Keywords:** leucine-rich repeat kinase 2 (LRRK2), Parkinson’s disease, striatum, synapse, cell-type specificity, protein kinase A (PKA), A-kinase adaptor protein (AKAP)

## Abstract

Mutations in leucine-rich repeat kinase 2 (LRRK2) cause Parkinson’s disease with a similar clinical presentation and progression to idiopathic Parkinson’s disease, and common variation is linked to disease risk. Recapitulation of the genotype in rodent models causes abnormal dopamine release and increases the susceptibility of dopaminergic neurons to insults, making LRRK2 a valuable model for understanding the pathobiology of Parkinson’s disease. It is also a promising druggable target with targeted therapies currently in development. LRRK2 mRNA and protein expression in the brain is highly variable across regions and cellular identities. A growing body of work has demonstrated that pathogenic LRRK2 mutations disrupt striatal synapses before the onset of overt neurodegeneration. Several substrates and interactors of LRRK2 have been identified to potentially mediate these pre-neurodegenerative changes in a cell-type-specific manner. This review discusses the effects of pathogenic LRRK2 mutations in striatal neurons, including cell-type-specific and pathway-specific alterations. It also highlights several LRRK2 effectors that could mediate the alterations to striatal function, including Rabs and protein kinase A. The lessons learned from improving our understanding of the pathogenic effects of LRRK2 mutations in striatal neurons will be applicable to both dissecting the cell-type specificity of LRRK2 function in the transcriptionally diverse subtypes of dopaminergic neurons and also increasing our understanding of basal ganglia development and biology. Finally, it will inform the development of therapeutics for Parkinson’s disease.

## 1. LRRK2 and Parkinson’s Disease

Parkinson’s disease (PD) is a progressive neurodegenerative disorder clinically characterized by motor symptoms including resting tremor, bradykinesia, and postural and gait instability [1]. These core symptoms are accompanied by sensory, cognitive, affective, and autonomic dysfunctions that may appear before motor symptom onset and increase in severity as the disease progresses [2]. Traditionally, genetics was thought to have a minor contribution to PD because of its low heritability. However, the last two decades have witnessed significant progress in understanding PD genetics [3]. While the vast majority of PD cases are sporadic, it is now clear that about 10% of all PD cases are monogenic forms of the disease with a clear family history [4,5]. The most common cause of familial PD is mutations to leucine-rich repeat kinase 2 (*LRRK2*), which leads to autosomal dominant, late-onset, progressive PD [6,7]. Disease-causing *LRRK2* variants account for up to 1–2% of sporadic PD and 5% of hereditary PD globally [8,9]. Although rare globally, the most common mutation, G2019S, is common among North African Arabs [10] and Ashkenazi Jews [11], accounting for 39% and 23% of total PD cases in these populations, respectively [8]. The mutations in what are considered to be sporadic cases likely reflect incomplete penetrance. Unlike other forms of monogenetic PD, the age of onset, clinical and neuropathological manifestations, and response to treatment are similar to idiopathic PD [8,12,13]. Intriguingly for a monogenetic disorder, the histopathology of *LRRK2*-PD cases is pleiomorphic, with variation even in closely related familial cases [14], reflecting the spectrum of histopathological phenotypes seen in idiopathic PD [15]. Although most G2019S mutation carriers have been associated with either brainstem-localized, transitional, or diffuse Lewy body pathology, not all individuals manifest Lewy pathology [16,17]. Other G2019S cases have been found to have neurofibrillary tangles, senile-plaques, Alzheimer’s Disease–like pathology, ubiquitin-positive inclusions, or TDP-43 proteinopathy [18,19]. Other mutations have commonly shown nigral cell loss with no accompanying pathology or Lewy body or tau pathology in various anatomic distributions [16]. A few differences in *LRRK2*-PD patients’ clinical symptoms compared to idiopathic PD have been described; these include the tremor as the first manifesting symptom, slower motor symptoms progression [20], and, in general, less frequent typical non-motor symptoms including hyposmia, sleep disturbances, and mood changes [9,21].

Interestingly, there is no male predominance across *LRRK2*-PD mutation carriers, unlike what is typically seen in sporadic PD cases [22]. Nevertheless, the response to dopamine therapy in *LRRK2*-PD patients is similar to that of patients with sporadic PD [8,9,13]. Overall, neuropathologically and clinically, all *LRRK2*-PD cases are characterized by neuronal loss in the substantia nigra pars compacta (SNc) and are largely similar to idiopathic PD. Therefore, understanding how LRRK2 dysfunction leads to PD will broadly inform the pathophysiological basis and therapeutic strategies in PD. This is further supported by compelling evidence from genome-wide association studies showing that *LRRK2* variants also act as risk factors for sporadic disease [23]. Recent studies also show that endogenous LRRK2 kinase activity is commonly increased in patients with late-onset sporadic PD, suggesting that LRRK2-targeted treatments, currently in development (https://www.denalitherapeutics.com/pipeline, accessed on 20 October 2021), could be of therapeutic value independent of *LRRK2* mutations. Intriguingly, LRRK2 may also play a role in another neurodegenerative disorder. SNPs at a locus on chromosome 12 were found to improve outcomes in Progressive Supranuclear Palsy while also reducing the expression of LRRK2 [24,25]. This finding potentially broadens the relevance of LRRK2 to neurodegenerative disease.

Recent research has demonstrated that LRRK2 is functionally relevant in multiple cell types throughout the basal ganglia circuit. Across brain regions, and among cell types within regions, LRRK2 is differentially expressed, and LRRK2 mutation results in distinct functional deficits in different cell types. In order to draw attention to this emerging frontier of LRRK2 biology, we discuss cell-type specificity of LRRK2 expression in the basal ganglia, the functional impacts of LRRK2 mutation in striatal cells, and several effectors and interactors that could potentially mediate the cell-type specificity of LRRK2 function. 

## 2. The LRRK2 Protein and Disease Modeling

LRRK2 is a large (286 kDa) multidomain protein, one of the few in the proteome that combines both kinase and GTPase activity [26,27]. At its N-terminus are a series of scaffolding domains, including an armadillo domain, an ankyrin domain, and the namesake leucine-rich repeat domain. The catalytic core of the protein contains the Roc-COR and kinase domains, while toward the C-terminus there is a WD40 domain [28]. Recently, a structural map of the full-length LRRK2 using cryo-EM was published [29]. This study followed two elegant complimentary reports where several LRRK2 domains were structurally resolved with high resolution [30,31]. These groundbreaking studies provided essential insights into the function of this rather unusual kinase, while at the same time illuminating the complexity of the LRRK2 biology.

The two most common pathogenic mutations, G2019S and R1441C/G/H, are located in separate domains. G2019S is located in the kinase domain and increases LRRK2 kinase activity. The other, R1441C/G/H, is found among a cluster of rarer pathogenic mutations at the ROC domain and impacts the ability to dissociate from GDP, slowing GTPase activity [32]. Both mutations lead to the increased phosphorylation of substrates [33], suggesting that directly and indirectly elevated kinase activity may be a common disease-causing mechanism [34]. However, the mechanism by which mutations outside of the kinase activity lead to increased kinase activity remains unclear [35]. Despite significant knowledge gaps on the biology of the LRRK2 protein, targeting LRRK2 kinase activity has been a promising therapeutic avenue, and, as mentioned above, small molecule LRRK2 kinase inhibitors are currently in clinical development. In addition to the brain, LRRK2 is highly expressed in lung, kidney, and immune tissues [36], raising concerns about the potential off-target effects of LRRK2 inhibitors. However, recent work has somewhat alleviated this concern. In macaques, histological alterations to lung tissue caused by LRRK2 are not connected to deficits in lung function, are reversed upon cessation of treatment, and are not observed in doses toward the lower end of the dosing range with therapeutic potential [37]. Furthermore, an analysis of LRRK2 predicted loss-of-function variants in biobank samples found that approximately 1 in 500 humans is heterozygous for such a variant, with no apparent changes to lung, kidney, or liver function, nor any discernible effects on health or lifespan [38]. These studies have reinforced LRRK2 as an important target for therapeutic interventions for PD.

Knock-out or pathological mutations of LRRK2 or its homologs have been recapitulated in various model organisms, including *C. elegans,* drosophila, zebrafish, and rodents [39]. In contrast to mammals, drosophila and *C. elegans* possess only a single homolog of LRRK2, called dLrrk and Lrk-1, respectively. In rodents, the loci of the most common mutations, G2019S, and R1441G/C/H, are conserved, with 88% conservation of the amino acid sequence and each of the protein’s domains [40]. 

## 3. Cell-Type Specificity of LRRK2 Expression

The temporal pattern of LRRK2 expression in the rodent brain indicates a potential role in development [41,42]. *LRRK2* transcription is undetectable in embryonic mice and rats. mRNA is first detected in the first week after birth, after which it increases gradually until reaching a plateau at about 3–4 weeks, at which time the transcript levels and spatial expression pattern will be maintained through adulthood [43,44,45,46,47]. This correlates with the rate of synaptogenesis in the striatum during a window of heightened plasticity during which activity sculpts corticostriatal connectivity with long-lasting consequences [48]. As such, abnormalities to the developing striatum conferred by LRRK2 mutations have the potential for long-lasting changes to the assembly and function of the striatal network.

Anatomically, *LRRK2* mRNA is expressed at low levels across most of the brain, but it is specifically enriched in brain regions receiving dopaminergic input, including the cerebral cortex and striatum [43,49,50]. Semiquantitative in situ hybridization and quantification of transcripts by qPCR indicates that expression in mouse, rat, and human tissue is highest in the striatum, including the nucleus accumbens, and the cerebral cortex. Meanwhile, the substantia nigra, along with the ventral tegmental area (VTA) and the rest of the midbrain, has substantially lower levels of *LRRK2* mRNA [50,51,52,53,54,55]. *LRRK2* expression in the thalamus varies among the nuclei [52,53], while the globus pallidus has low or barely detectable *LRRK2* transcripts [52,53,54]. Complementing these data, single-cell transcriptomics data such as that presented at dropviz.org [56] has recently revealed details about *LRRK2* expression broken down by genetically defined cell type (Figure 1).

### 3.1. Striatum

#### 3.1.1. SPNs

SPNs make up about 90% of striatal neurons, with the remainder consisting of cholinergic interneurons and several distinct types of GABAergic interneurons. In both rodents and humans, LRRK2 protein is strongly expressed in SPNs of both the direct and indirect pathways, accounting for the strong LRRK2 expression observed in striatum [47,54,57]. There is considerable heterogeneity within the striatum, with LRRK2 expression being generally higher in striosomal dSPNs than in those inhabiting the matrix, and with generally higher expression in lateral compared to medial striatum [56,57]. During early postnatal development, the LRRK2 protein can first be detected around P8 in striosomes; expression is detectable in the matrix by P16 [57]. In addition to SPNs of the direct and indirect pathways, LRRK2 is strongly expressed in eccentric SPNs, which express both D1Rs and D2Rs [56]. As discussed in the following sections, despite similar expression in direct and indirect-pathway SPNs, the phenotypes manifested by LRRK2 mutation differ between the two pathways [58,59], likely due to interactions with differentially modulated signaling pathways, including PKA. PKA signaling is oppositely affected by dopamine in the direct and indirect pathways [60].

#### 3.1.2. GABAergic Interneurons

After SPNs, most of the remaining neurons of the striatum are constituted by a diverse variety of GABAergic interneurons. The most commonly used markers to distinguish between subtypes of striatal GABAergic interneurons are parvalbumin, calretinin, and nitric oxide synthase [61,62]. In human tissue, apparent LRRK2 expression has been observed in a majority of calretinin-expressing interneurons, as well as a small subset of parvalbumin- and nitric-oxide-synthase-expressing neurons [55]. However, in mice, LRRK2 staining was observed in most parvalbumin- and nitric-oxide-synthase-positive neurons and only a few calretinin-positive cells [54]. In contrast, other groups have found no colocalization between LRRK2 and any of these markers [57]. Single-cell mRNA expression across all types of GABAergic interneurons is approximately an order of magnitude lower than in SPNs [56]. If the expression of the LRRK2 protein correlates with the mRNA levels reported by Saunders and colleagues [56], striatal interneurons would contain much less protein than the surrounding SPNs. This would increase the difficulty of detection, potentially explaining part of the apparent discrepancy among these studies. 

#### 3.1.3. Cholinergic Interneurons

Immunohistochemical detection of LRRK2 is notoriously difficult, which has led to a plethora of reagents and methodologies, and, consequently, some conflicting results. In the best-validated experiment to date, LRRK2 immunoreactivity was found in a minority (about 8%) of cholinergic interneurons (CINs) [57]. Others, studying tissue from mice, rats, and humans, have reported that the majority of CINs express LRRK2 [54,55,63]. This implies consistency of LRRK2 expression in CINs across species. Meanwhile, as with GABAergic interneurons, the reports leave open the question of whether LRRK2 is expressed at low levels in most (or all) CINs, hampering detection, or whether it is expressed at higher levels in a smaller number of CINs as Mandemakers and colleagues suggest [57]. The levels of mRNA reported in CINs by Saunders and colleagues are slightly higher than in GABAergic interneurons and broadly similar to levels reported for various dopaminergic neuron subtypes [56]. 

CINs may contribute to basal ganglia abnormalities arising from LRRK2 mutations through a specific vulnerability to LRRK2 mutation. In mice carrying a germline R1441C LRRK2 mutation, CINs, but not other types of striatal neurons, lack primary cilia [64]. Phosphorylation of the LRRK2 substrate Rab10 impairs primary cilia formation [64,65]. Among other functions, primary cilia serve as specialized structures for the sonic hedgehog (Shh) signaling reception. Shh released from dopaminergic and striatal neurons stimulates the release of trophic factors that provide essential support to dopaminergic axons [65,66]. Thus, the maintenance of primary cilia represents a functional role for LRRK2 that is both specific to cholinergic neurons and mechanistically linked to the integrity of dopaminergic axons.

### 3.2. Striatal Afferents

#### 3.2.1. Cerebral Cortex

Along with striatum, the cortex has some of the highest levels of LRRK2 mRNA and protein expression [50,51,52,53,54,55]. LRRK2 is expressed throughout the cerebral cortex and in principal neurons of all layers [52,53,54,55,57]. However, reports have varied as to whether expression is largely uniform [53] or varied by region and cortical layer. Variations that have been reported include: sparser mRNA expression in layers III and IV of murine somatosensory and parietal cortex [52]; enhanced protein expression in layers II/III and superficial layer V [54]; and reduced protein in layer VI [57]. In both rodent and human cortex, the principal neurons of each layer account for the majority of the LRRK2 signal, but some parvalbumin-, nNOS-, or calretinin-positive interneurons also express LRRK2 [54,55]. Consistent with this ISH and IHC data, with single-cell RNA-seq, *LRRK2* expression can be found across the variety of cell types in the cortex, with pyramidal neurons generally appearing to express more LRRK2 mRNA than most types of GABAergic interneurons, but with significant variation across subtypes [56].

#### 3.2.2. Thalamus

In addition to the cerebral cortex, the thalamus sends glutamatergic projections to the striatum. *LRRK2* mRNA can be observed throughout the thalamus, but mRNA expression is relatively higher in the paraventricular and reticular nuclei [52,53]. 

#### 3.2.3. Dopaminergic Midbrain

As with the rest of the brain, *LRRK2* mRNA is first detectable in the dopaminergic midbrain in the first postnatal week; however, expression plateaus earlier than in the cortex and striatum [44,45]. *LRRK2* mRNA is present in both the dorsal and ventral tiers of the SNc, as well as in the ventral tegmental area (VTA), while weaker expression can be detected in the pars reticulata (SNr) [52]. In both rodent and human brain, *LRRK2* mRNA and protein colocalize with neuromelanin and tyrosine hydroxylase [54,55].

**Figure 1 cells-11-00169-f001:**
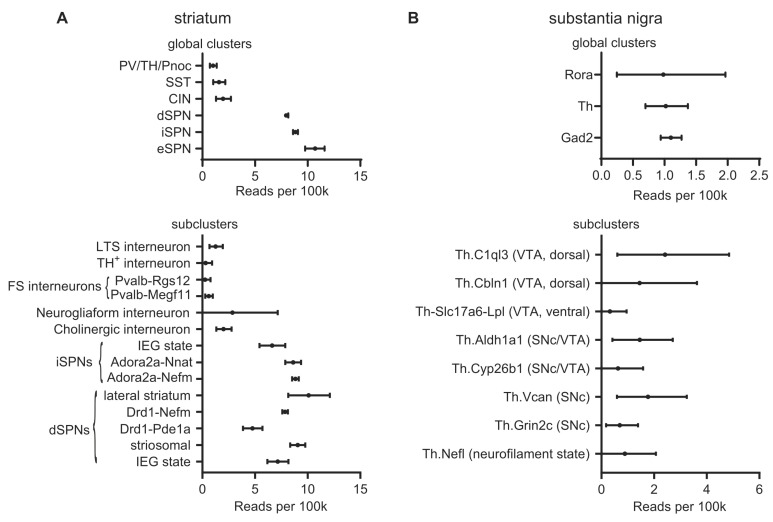
Single-cell LRRK2 mRNA levels. Single-cell LRRK2 mRNA expression data from dropviz.org for selected types of striatal neurons and dopaminergic midbrain projection neurons. (**A**) Top: LRRK2 mRNA expression in global clusters defining striatal neuron types. LRRK2 mRNA is more abundant in SPNs than in striatal interneurons. Bottom: Subclusters selected on the basis of their putative identity with well-characterized subpopulations of striatal neurons (i.e., excluding eSPNs and clusters which had been identified as non-striatal neurons). Cell cluster identities are as described by Saunders et al., except Neurogliaform interneurons, which was assigned based on the expression of NPY but not parvalbumin, somatostatin, or Nos1, as defined by Tepper et al. [62]. Some subclusters of fast-spiking interneurons and SPNs do not correspond to a well-characterized cell type and are referred to by the subcluster name. Consistent with immunohistochemical evidence by Mandemakers et al. [57], LRRK2 mRNA is elevated in lateral and striosomal dSPNs. These data can be accessed in the context of the full dataset at [http://dropviz.org/?stateid=8afe26552f34f746] (accessed on 20 October 2021). (**B**) Top: LRRK2 mRNA expression in global clusters defining neuron types in the substantia nigra. Bottom: LRRK2 mRNA levels in the subclusters constituting the “Th” (Tyrosine hydroxylase-expressing) global cluster, i.e., dopaminergic neurons of the SNc and VTA. Cell types are as defined by Saunders et al. LRRK2 mRNA expression is low, but detectable, across dopaminergic subtypes. These data can be accessed in the context of the full dataset at [http://dropviz.org/?stateid=8c997a178c82063c] (accessed on 20 October 2021). Abbreviations: PV: parvalbumin; TH: tyrosine hydroxylase; Pnoc: prepronociceptin; SST: somatostatin; CIN: cholinergic interneuron; eSPN: eccentric SPN [56]; LTS: low-threshold spiking; FS: fast-spiking; IEG: immediate early gene; Rora: RAR-related orphan receptor A; Gad2: glutamic acid decarboxylase 2.

## 4. Modulation of Synaptic Function by LRRK2 Mutants

### 4.1. Divergent Effects of LRRK2 Mutation at Dopaminergic and Glutamatergic Terminals

Several LRRK2 mutant animals exhibit deficits in dopamine transmission and dopamine-responsive behaviors, suggesting that synaptic dysfunction might represent a crucial pathological feature of LRRK2. Rodents with LRRK2 knocked out do not exhibit overt Parkinsonian phenotypes such as synuclein pathology, loss of dopaminergic neurons, or motor dysfunction [67,68,69,70,71]. In contrast, the overexpression of wild-type or mutant LRRK2 by bacterial artificial chromosomes or cDNA can promote neurodegeneration. Subsequent work carried out in knock-in models, in which the transgene is expressed at more physiological levels, has revealed several subtler phenotypes [72,73]. Although these models do not display an overt loss of dopaminergic neurons, knock-in of LRRK2 pathogenic mutations does result in age-dependent reductions in dopaminergic tone and evoked dopamine release, along with histopathological manifestations including mitochondrial abnormalities and elevated tau [73,74,75]. Intriguingly, phenotypes in these models are not restricted to dopaminergic neurons. In contrast to the reduced dopaminergic tone and dopamine transmission, several studies have found enhanced glutamatergic transmission [47,74,76,77]. A recent study showed that a G2019S-mediated slowing of endocytosis was observed specifically in primary dopamine neurons and not in cortical or hippocampal neurons, suggesting a potential mechanism underlying selective vulnerability in PD [77]. It is unclear what accounts for the differential impact of LRRK2 on neurotransmitter release from these distinct classes of neurons. Additionally, it remains to be determined whether an LRRK2 mutation differentially affects the presynaptic function of different midbrain neuromodulatory projection neurons or even different dopaminergic neuron subtypes. 

### 4.2. Functional Impacts on Glutamatergic Striatal Afferents

In addition to the phenotypes observed at dopaminergic neurons, LRRK2 knock-in models reveal altered structure, function, and plasticity at striatal synapses [59,74,76,78,79,80]. This has led to increasing recognition that LRRK2 is functionally relevant in the basal ganglia prior to old age and in a context that may be distinct from neurodegeneration; in addition, it may contribute to neurodegeneration later in life [41,49]. As previously discussed, LRRK2 expression in the cortex and striatum increases gradually, starting in the first postnatal week through 21 days of age [41,42,43,46,47]. This correlates with the rate of synaptogenesis in the striatum during a window of heightened plasticity in which activity sculpts corticostriatal connectivity with long-lasting consequences [41,42,48]. As such, abnormalities to the developing striatum conferred by LRRK2 mutations have the potential for long-lasting changes to the assembly and function of the striatal network. Early indications of a glutamatergic synaptic phenotype in neurons with LRRK2 mutations come from studies carried out in primary culture. In primary cultures of cortical neurons overexpressing various LRRK2 constructs, wild-type or kinase-dead LRRK2 overexpression does not change synaptic transmission. However, transfected primary neurons overexpressing G2019S or R1441C LRRK2 mutations have more dendritic protrusions, increased glutamatergic synaptic input, and larger AMPA and NMDA currents after 17 days in vitro [81]. Meanwhile, primary neurons with a knock-in G2019S mutation are morphologically normal, but have an increased frequency of miniature excitatory postsynaptic currents (mEPSCs) at 21 days [47]. The age of the cells is meaningful since knockout and mutant forms of LRRK2 can cause a transient dendrite outgrowth phenotype that affects synaptic connectivity, but is largely normalized by 21 days in vitro [46,47]. However, other factors such as the degree of mutant LRRK2 overexpression could also account for the apparent inconsistency. 

At corticostriatal synapses, consistent with the results from primary cultures, the overexpression of wild-type LRRK2 or knockout of endogenous LRRK2 does not impact basal spontaneous neurotransmission in the dorsal striatum of young adult mice [80], but the expression of PD-associated mutations does. G2019S knock-in mice have differences in synaptic activity that vary throughout their development. The frequency of miniature and spontaneous EPSCs onto SPNs in the dorsal striatum of these mice is transiently increased during a period from 3 weeks to 3 months of age [74,76]. This period coincides with LRRK2 peaking after gradually rising during the preceding weeks [41,42,43,47]. Both before and following this temporal window, the synaptic input appears normal, suggesting that other factors compensate for the LRRK2-dependent changes [74,76]. The increased synaptic activity is likely due to an increased drive from cortical projection neurons, since the density of SPNs’ synapses and dendrites were unchanged and because the increased EPSC frequency could be normalized by severing the cortex from the striatum [76]. The mechanisms driving increased cortical activity have not been further explored. SPNs also had enlarged spine heads and a correspondingly broadened distribution of amplitudes of spontaneous synaptic events [76]. These changes observed in the dorsal striatum are distinct from the alterations in the nucleus accumbens of the same G2019S knock-in animals, where at P21, the frequency of excitatory synaptic events is unaltered, despite similarly increased event amplitude and correspondingly enlarged spine heads [79]. As in the dorsal striatum, these synapses appear to normalize as mice reach adulthood (by 10–12 weeks). However, a social defeat paradigm can reveal apparent underlying differences which cause an increased frequency and amplitude of sEPSCs onto SPNs of the nucleus accumbens in G2019S knock-in, but not in wild-type, animals [79].

Glutamate release from corticostriatal afferents is negatively modulated by axonal D2 receptors. The D2 receptor agonist quinpirole decreased the frequency of spontaneous synaptic currents and the amplitude of evoked currents in G2019S knock-in mice to a greater degree than mice with wild-type, kinase-dead, or knocked-out LRRK2 [82]. Paired pulse facilitation, in which a closely preceding first stimulus potentiates a second stimulus, is reduced at corticostriatal synapses in G2019S knock-in mice, but only when the stimulating electrode is placed within the striatum, where it can also stimulate the release of dopamine and other striatal neuromodulators [80]. Thus, G2019S disrupts the regulation of release probability by dopamine at corticostriatal synapses. This could be rescued by the D2R antagonist remproxide, specifically in axons projecting onto dSPNs [80]. 

### 4.3. Impact of Mutant LRRK2 at Postsynaptic Sites in SPNs

The existence of enlarged spine heads in SPNs raises the question of whether LRRK2 mutations can abnormally potentiate synapses. Any pathway-specific effects of LRRK2 mutation are particularly relevant to PD pathology since there is a well-established pathway-specific difference in morphological and physiological changes to SPNs after dopamine depletion [83,84]. Our laboratory recently performed a comparative analysis of two PD-related LRRK2 mutations in direct- and indirect-pathway SPNs. Employing global- and single-synapse approaches, we found that LRRK2 mutations differentially impact the structure and function of dendritic spines of these two cell types (Figure 2). In R1441C knock-in SPNs, the postsynaptic density (PSD) fractions of striatal synaptosomes had increased GluA1 content, along with increased phosphorylated PKA and increased phosphorylation of several PKA substrates, including GluA1 [78]. Analysis of dendritic spines using super-resolution microscopy revealed increased GluA1 colocalization with PSD95, specifically in direct-pathway SPNs. This was confirmed by using laser light to uncage glutamate near individual spines, generating larger single-synapse EPSCs in dSPNs, but not iSPNs. In contrast, G2019S knock-in mice did not have increased GluA1 content in synaptosomes [59,78], or uncaging evoked EPSCs in either SPN subpopulation. While they did have increased colocalization between GluA1 and PSD95 in dendritic spine heads, this was specific to indirect-pathway SPNs [78]. The synaptic changes did not translate into an increase in the average amplitude of spontaneous synaptic events in either SPN subpopulation. Other work also shows that sEPSC amplitude in dSPNs and iSPNs are not differentially impacted by the G2019S mutation. However, the distribution of sEPSC amplitudes is broadened or shifted toward larger amplitudes [74,78,80,82]. Presynaptic effects, such as the aforementioned differential modulation of release probability by G2019S, could potentially explain the apparent discrepancy between enhanced uncaging-evoked EPSCs and minimally altered sEPSC amplitudes.

Despite the abnormal increase in synaptic AMPARs in R1441C dSPNs, experiments interrogating the ratio of AMPAR to NMDAR inserted into the SPN postsynaptic membrane have not shown such an increase in the LRRK2 knockout, overexpressing, or G2019S knock-in mice [74,80,82]. SPNs may be less vulnerable in this regard to G2019S and its increased kinase activity than they are to additional effects of mutations to R1441, which can potentially impact cell biology in a variety of ways beyond the indirect increase in kinase activity. Accordingly, intrinsic excitability is decreased in indirect-pathway SPNs in R1441C knock-in mice, but not in either subpopulation in G2019S mice [75,82]. Alternatively, at hippocampal Schaeffer collateral synapses, G2019S LRRK2-dependent increases to the AMPAR/NMDAR ratio are sensitive to dialysis from the patch electrode. They are only observed recording through a perforated patch configuration [85], raising the possibility that the specific methodology used could be similarly masking AMPAR increases at corticostriatal synapses, or that hippocampal synapses may be vulnerable to LRRK2 kinase activity in a way that corticostriatal synapses are not. Whether or not the G2019S mutation alters total AMPARs inserted into the synapse, it alters the subunit composition of striatal AMPAR receptors. G2019S knock-in mice have reduced calcium permeability of AMPARs, revealed by the loss of inward rectification at depolarized membrane potentials and insensitivity to NASPM, an antagonist specific to calcium-permeable AMPARs [59]. This effect was observed in both direct- and indirect-pathway SPNs.

The abnormalities in AMPAR receptors at synapses with LRRK2-associated mutations raise questions about these synapses’ ability to express plasticity. High-frequency stimulation of corticostriatal afferents can evoke LTD through a mechanism involving retrograde endocannabinoid signaling from the postsynaptic neuron to reduce glutamate release, while LTP can be evoked by high-frequency stimulation paired with depolarization of the postsynaptic neuron or low extracellular magnesium [86]. In the nucleus accumbens in G2019S knock-in mice, an NMDAR-dependent LTP induction protocol, in which cells were depolarized while afferents were stimulated at 0.1 Hz, failed to potentiate synapses onto both dSPNs and iSPNs [59]. Moreover, there were pathway-specific effects; synapse strength on dSPNs returned to baseline after the induction protocol, while iSPNs expressed LTD instead of LTP. This is in contrast with experiments in the dorsal striatum of BAC-G2019S overexpressing mice, which have found that corticostriatal LTP induced by high-frequency stimulation under low-magnesium conditions was intact; however, in these experiments, LTD induced by high-frequency stimulation in normal-magnesium conditions was impaired [87]. These changes in corticostriatal plasticity correlate with specific learning impairments: G2019S knock-in mice were more resilient than WT mice to chronic social defeat stress carried out over several days [59]. Following the stress paradigm, susceptible wild-type mice had increased calcium-permeable AMPARs in the nucleus accumbens, while G2019S and the resilient wild-type mice did not. Meanwhile, these mice show increased social avoidance and sucrose consumption after acute social defeat stress [79]. Mice overexpressing wild-type LRRK2 have impaired long-term, but not short-term, novel object recognition [80]. 

### 4.4. Non-Striatal Synapses

Abnormalities in the basal transmission and plasticity at glutamatergic synapses are not confined to the striatum. At hippocampal Schaffer collateral synapses, the overexpression of G2019S LRRK2 in a BAC transgenic mouse line causes increased basal evoked synaptic transmission and an increased AMPAR/NMDAR ratio [85]. Although multiple forms of short- and long-term synaptic plasticity were expressed typically at these synapses, including post-tetanic potentiation and high-frequency stimulation LTP, a low-frequency stimulation protocol failed to produce LTD in the transgenic mice, except in the presence of a LRRK2 kinase inhibitor. Another research group, using a different transgenic mouse line to overexpress G2019S LRRK2, was able to elicit LTD using low-frequency stimulation at Schaeffer collateral synapses under different experimental conditions [87]. These findings indicate that hyperactive LRRK2 kinase activity is unlikely to abolish LTD at these synapses altogether. 

Collectively, evidence from G201S knock-in models reveals that pathologically increased LRRK2 kinase activity impairs multiple forms of plasticity across various cell types, although not universally. A systematic dissection of the specific circumstances under which LRRK2 mutations impair corticostriatal LTP and LTD would be required to make more substantive conclusions. For example, in SPNs, LTP induced by high-frequency stimulation or theta-burst stimulation requires activation of D1Rs or A2ARs and the resultant PKA activity [88], while in the hippocampus, the requirement for PKA depends on the temporal spacing of the stimulus [89,90]. Therefore, a systematic comparison of mechanistically distinct forms of plasticity in animals carrying LRRK2 mutations could provide mechanistic insights into the synaptic functions of LRRK2.

## 5. LRRK2 Postsynaptic Mechanisms

### 5.1. PKA

PKA is the main effector of dopamine signaling in the SPNs. Opposing effects of dopamine receptor activation on PKA activity in direct- and indirect-pathway SPNs is fundamental to the function of the basal ganglia circuit [60]. The PKA holoenzyme consists of two catalytic (PKAC) and two regulatory subunits. The last can be subdivided into types I and II [91]. The PKARIIβ regulatory subunit is particularly abundant in the SPNs, and phenotypes of the RIIβ knockout mice are mainly related to the striatal functions, as they display severely defective PKA activity and defective responses in two experience-dependent locomotor behaviors: the rotarod task and amphetamine-induced sensitization [92]. There is an exciting and somewhat complicated cross-talk between LRRK2 and PKA in striatal neurons. First, LRRK2 regulates the localization of PKA by an interaction between its ROC domain and a pool of PKA through its direct binding with the PKARIIβ subunit in the dendritic shaft. By this mechanism, LRRK2 regulates the effects of PKA by functioning similarly to a dendritic AKAP (A-kinase adaptor protein) [46]. LRRK2 knockout impairs the phosphorylation of cofilin by PKA, which is important for cytoskeletal remodeling and plasticity, and R1441C knock-in mice have an increased cofilin phosphorylation and GluR1 [46], as well as an increased phosphorylation of PKA substrates in synaptosomal fractions [78].

Specificity in fundamental striatal functions seems incompatible with the diffusion of molecules of the PKA pathway, suggesting that mechanisms are required to produce local cAMP activation [93]; this can be achieved, at least in part, from the compartmentalization of PKA enzymes in neurons. It is now clear that postsynaptic PKA is confined to various subcellular compartments by anchoring molecules such as AKAPs [94]. Based on these and our previous findings [46,78], we have formulated the following working model: the R1441C pathogenic mutation found on the ROC domain—the LRRK2 domain, which interacts with PKARIIβ—results in decreased LRRK2-PKA binding. This, in turn, leads to an increased translocation of the otherwise dendritic-based PKA pool into the dendritic spines. This likely leads to more PKA bound to synaptic AKAPs, such as AKAP5, bringing the PKA holoenzyme closer to the upstream components of the cAMP signaling pathway, such as the dopamine receptors and adenylyl cyclase (AC), following dopamine stimulation. This altered localization may account for the increased phosphorylation of synaptic targets we observed in R1441C KI striatal extracts [46,78]. Accordingly, previous studies showed that neurons lacking AKAP5, or expressing a PKA anchoring-deficient AKAP5, exhibit a significant translocation of the synaptic PKARIIβ subunits to the dendritic shafts, where they bind to the dendritic AKAP MAP2 [95,96]. Furthermore, in *MAP2Δ1-157* mice, in which the PKA binding site of MAP2 was genetically deleted [97], a redistribution of PKARIIβ in the dendritic spines is observed. This is an example of how the impaired binding of PKARIIβ to AKAPs in specific subcellular compartments results in altered PKARIIβ localization. 

Since our previous findings suggest that LRRK2 acts as a dendritic shaft AKAP and regulates PKA by directing its subcellular localization, we were interested in identifying candidate amphipathic helices in the ROC domain of LRRK2 that could serve as anchors for PKA. AKAPs are diverse in overall structure, yet share the functional ability to anchor PKA through an amphipathic helix. We used the online HeliQuest server (http://heliquest.ipmc.cnrs.fr/, accessed on 4 January 2018) with an 18 amino acid window input to identify candidate helices within the ROC domain as amphipathic according to the default parameters defined by the program, including the hydrophobicity and hydrophobic moment. In our analysis we ruled out lipid-binding helices or transmembrane segments. An interesting candidate helix emerged corresponding to the sequence “AEVDAMKPWLFNIKAR” within the ROC domain (Figure 3). Further inspection confirmed this sequence exhibits predicted characteristics of an amphipathic helix and structurally was identified correctly as a helix. Of particular interest, the arginine at the C-terminus of this helix is a known mutation in clinical cases (R1441). We used the recently determined cryo-EM structure (PDB entry 7LI3) to visualize the environment around R1441 [29] and found R1441 nestled between the backbone carbonyls of Met1409 and Trp1791 from the ROC and COR domains, respectively. A comparison of the full LRRK2 structure with a cryo-EM sub structure lacking the scaffolding domains reveals a shift in the orientation of the ROC and COR domain interface with the R1441 unseated from Met1409, yet still in interaction with Trp1791 [30]. The Met1409 of the structure of LRRK2 without the scaffolding domains is not present in the sub structure model. While the initial predictions of the amphipathic helix containing R1441 cannot tell the interaction partners, the molecular structures suggest a dynamic environment for R1441 with neighboring residues and domains. Although this is an exciting finding, further experiments are needed to help clarify the details of this intriguing mechanism. 

PKA may also act as an upstream regulator of LRRK2 by phosphorylating it at two sets of serine residues, 910 and 935, which are between the ankyrin and leucine-rich repeat domains, and S1443 and S1444, which are located in the Roc GTPase domain [98,99,100,101]. However, in direct opposition to these findings, other investigators have found that treating cells with the PKA activator forskolin, along with the phosphodiesterase inhibitor IBMX, does not increase S935 phosphorylation in cells. Further, that treatment with the PKA inhibitor H-89 does not reduce S935 phosphorylation [102,103,104], casting doubt on whether PKA does in fact act as a kinase toward LRRK2. Two of these residues, S935 and S1443, are critical for binding to 14-3-3, which stabilizes LRRK2 in a conformation that renders its kinase domain inactive and regulates its association with different cellular compartments [105]. Both the phosphorylation of LRRK2 by PKA and the AKAP-like function of LRRK2 are sensitive to GTP binding and the R1441C/G/H mutations in the ROC domain. In addition to the effect of R1441C mutation on the PKA localization, GTP binding and all three PD-associated mutations at R1441C/G/H prevent PKA from phosphorylating the S1443/4 residues of LRRK2 [100]. Thus, the reduced interaction with 14-3-3 is, in theory, permissive of increased kinase activity (although the degree of increased kinase activity in these mutants is controversial, given conflicting results in the literature).

Finally, a third mode of interaction has been described in microglia, though not in neurons. In microglia, LRRK2 kinase activity promotes increased phosphodiesterase activity, acting as an upstream negative regulator of PKA [106]. With its N-terminal region composed of several scaffolding domains, there is an implied different function for LRRK2 as a structural protein that regulates other signaling pathways, similar to its mode of regulation for PKA. Another such interactor is the regulator of protein transport from the endoplasmic reticulum sec16a [107]. Sec16a binds to the LRRK2 ROC domain; through this interaction, sec16a is anchored at dendritic endoplasmic reticulum exit sites (ERES). LRRK2 loss-of-function leads to sec16a being diffusely distributed throughout the cytoplasm and impaired ERES function, closely phenocopying the effects of sec16a knockout. In dendrites, this impaired ERES function reduces the export and membrane insertion of multiple NMDAR subunits [107].

### 5.2. Rabs

Much attention has been directed toward the recent discovery of multiple Rabs as LRRK2 substrates. Rabs are small GTPases. In their active, GTP-bound form, Rabs associate with membranes where they recruit other proteins, define the identity of membranous structures, and regulate processes such as trafficking and vesicle fusion. Rabs play an important role in the maintenance and plasticity of synapses by regulating the trafficking of receptors from the Golgi apparatus along the dendrite and into dendritic spines, the exocytosis and endocytosis of receptors during plasticity, and the recycling or degradation of endocytosed receptors [108,109]. Two Rabs with roles at the postsynapse, Rab5 and Rab8, have been identified as substrates of LRRK2 [33,110]. In vitro, Rab8 is phosphorylated by LRRK2 at a rate 10-fold greater than another in vitro LRRK2 substrate, moesin [33]. Rab8 can be found in dendritic spines near the postsynaptic density, where it regulates the trafficking of AMPAR-containing exocytic vesicles between the Golgi complex and the membrane [111,112]. In organotypic hippocampal slices, blocking Rab8 activity by overexpressing dominant-negative GDP-bound Rab8 mutant (T22N) results in smaller AMPAR currents, smaller dendritic spines, an accumulation of AMPARs inside the spine head, and an inability to express long-term potentiation [111]. While Rab8 is required for the exocytosis of newly synthesized AMPARs, maintaining AMPAR-containing synapses also requires the recycling of endocytosed AMPARs by pathways that are impaired by dominant negative GTP-bound forms of other Rabs, including Rab4 and Rab5 [113]. As with de novo AMPARs, membrane targeting of GABA_A_ receptor subunits is also impaired by T22N Rab8 [113]. However, these experiments should be interpreted with caution. The phosphorylated fraction of cellular Rab is small, being only about 1% [33]. Although the exogenous expression of GDP-bound Rabs increases the GDP-bound portion of the endogenous pool, it is unclear exactly which portions of the pool are affected. Increased phosphorylation of Rab8 has been observed in synaptosomes prepared from the striatum of R1441C knock-in mice, placing it in a position to affect AMPAR trafficking in neurons with abnormal levels of AMPARs in the synaptic membrane [78].

Meanwhile, the endocytosis of AMPARs depends on Rab5. Unlike Rab8, LRRK2 phosphorylation of Rab5 has only been conclusively demonstrated in vitro, not in live cells [110,114]. Nevertheless, LRRK2 and Rab5 interact in cultured primary neurons, where they colocalize at endocytic vesicles and regulate endocytosis at synaptic terminals [115]. In hippocampal slice cultures, Rab5 is abundant near the postsynaptic membrane adjacent to the PSD. Overexpressing dominant-negative GDP-bound mutant Rab5 (S34N) blocks LTP, while overexpressing wild-type Rab5 mimics LTD by leading to AMPAR internalization, weakening AMPAR currents, and preventing further synaptic depression [116]. Rab5 function is necessary for AMPAR internalization downstream of NMDAR-mediated and mGluR-mediated LTD [117,118,119]. Specifically, overexpressing GDP-bound Rab5 impacts the pool of GluR1- and GluR2-containing AMPARs involved in synaptic plasticity while constitutively cycled GluR3-containing AMPARs, along with NMDARs and GABA_A_Rs, are minimally impacted [113,116]. How, if at all, LRRK2 phosphorylation of Rab5 and Rab8 affects their functions at postsynaptic sites has not been directly explored. The interaction between LRRK2 and Rab5 at postsynaptic sites may be similar to these two proteins at endocytic vesicles in axon terminals. Meanwhile, LRRK2 phosphorylation of Rab8 has been studied mechanistically and functionally in the context of ciliogenesis, where the phosphorylation of Rab8 or Rab10 at a conserved residue regulates their binding with the effector proteins RILPL1 and RILPL2, and thereby the formation and maintenance of cilia [64,65,114]. Presumably, these, or other effector proteins, could interact with Rab8 to regulate the delivery of AMPAR receptors into the synapse, an attractive potential explanation for the abnormalities at glutamatergic synapses observed in LRRK2 mutant models. 

### 5.3. PPM1H

The previously poorly characterized protein phosphatase PPM1H has recently been revealed to antagonize the kinase activity of LRRK2 by selectively dephosphorylating Rabs [120]. PPM1H was uncovered in a screen for pS73-Rab10 phosphatases, and follow-up experiments showed phosphatase activity against the equivalent site on the LRRK2 substrates Rabs 8A, 8B, 10, and 35 [120]. The closely related protein phosphatase PPM1M suppressed phosphorylation of Rabs as well, albeit to a much lesser degree, but is expressed at very low levels in the brain [120]. Another close relative, PPM1J, lacks a flap domain that mediates the interaction between PPM1H and its Rab substrates, preventing it from acting on the group of Rabs mentioned above [121]. Meanwhile, PPM1H also dephosphorylates Rab7a, a substrate of LRRK1, but not LRRK2 [122]. As evidence for a functional role of PPM1H antagonizing LRRK2 phosphatase activity, a PPM1H knockout increases the endogenous phosphorylation of Rabs in cultured lung epithelial cells, and phenocopies the ciliary deficit caused by LRRK2 mutation in cultured murine embryonic fibroblasts as well as in vivo in striatal CINs and astrocytes [65,120]. Differential expression or activity of PPM1H across striatal cell types has not been investigated in detail. However, mRNA copy number varies by cell type in a pattern that is largely opposite to that of LRRK2: copy number is higher in GABAergic interneurons of the cerebral cortex and striatum than in dopaminergic neurons of the SNc, striatal CINs, and cortical projection neurons, and it is lower still in striatal SPNs [56]. PPM1H has also appeared on a screen of transcripts regulated in neuronal plasticity, downregulated in primary hippocampal cultures during NMDAR-dependent chemical LTP [123], hinting at dynamic regulation and a potential role in synapse remodeling.

### 5.4. Calcium Homeostasis

Links between LRRK2 and calcium homeostasis have been investigated functionally with regard to apoptosis, neurite outgrowth and retraction, mitochondrial function, and autophagy. At the synapse, calcium enters the cell through NMDA receptors and Ca_V_1.2 and 1.3 channels located in the dendrite and perisynaptic region, activating calcium sensors such as CaMKII and calcineurin. These, in turn, regulate the activity and membrane targeting of synaptic proteins and can initiate the local protein translation and gene transcription required to express long-term synaptic plasticity. Data from calcium imaging experiments in cultured mouse and human iPSC-derived neurons have shown that LRRK2 kinase activity alters cytosolic calcium dynamics, with a sustained increase in cytosolic [Ca^2+^] during depolarization [124,125,126,127]. Patient-derived neurons also have altered gene expression profiles consistent with elevated cytosolic calcium [127]. Calcium enters the cytoplasm from either the extracellular space through the voltage- or ligand-gated calcium channels or intracellular compartments, primarily the endoplasmic reticulum, and is buffered in the cytoplasm by calcium-binding proteins, as well as by sequestration into organelles including the endoplasmic reticulum. Several phenotypes of cultured cells expressing PD-associated mutant forms of LRRK2, including reduced neurite length, increased numbers of autophagosomes, and mitochondrial trafficking defects, can be rescued by both extracellular and intracellular calcium chelation [124,126,128]. Neurite length and a mitochondrial trafficking defect can be rescued by nitredipine, an L-type calcium channel blocker, in neurons transfected with either GS or RC LRRK2 [124]. In HEK cells exogenously expressing Ca_V_2.1, and in PC12 cells, which have endogenously expressed voltage-gated calcium channels, overexpressing the WT and GS forms of LRRK2 increased the amplitude of calcium currents, an effect blocked by the LRRK2 kinase inhibitor MLi-2 [129]. In tissue from mouse brain, LRRK2 co-immunoprecipitated with the auxiliary modulatory voltage-gated calcium channel subunit Ca_V_β, demonstrating an interaction in vivo. Collectively, LRRK2 interacts directly with voltage-gated calcium channels, and LRRK2 kinase activity increases voltage-gated calcium currents.

Meanwhile, the intracellular ER-localized calcium sensor, Cepia-ER, reveals reduced calcium in the ER of dopamine neurons derived from human IPSCs expressing G2019S LRRK2, along with the reduced expression of several mRNAs associated with ER calcium handling [126]. Additionally, LRRK2 plays a role in regulating the translation of voltage-gated calcium channels, along with several other genes implicated in calcium homeostasis [127]. The mRNAs of voltage-gated calcium channels have diverse secondary structures, with many having an unusually complex 5′ UTR, a feature that would typically reduce translation. However, LRRK2 kinase activity disproportionately increases the translation rate of mRNAs with complex 5′ UTRs, contributing to an increased translation efficiency for multiple subunits of L-type voltage-gated calcium channels [127]. Patient-derived neurons expressing the G2019S mutation can be rescued by preventing LRRK2 phosphorylation of the s15 ribosomal subunit. Phosphorylation of s15 by LRRK2 had been previously linked to an increase in total protein synthesis [130]. Local protein translation at the synapse is mechanistically linked to long-term plasticity and neurodevelopmental disorders including ASD, fragile X syndrome, and tuberous sclerosis [131], raising the additional question of which, if any, locally translated proteins may be similarly regulated by LRRK2. The LRRK2 effectors mentioned here are summarized in Figure 4. 

## 6. Conclusions

Since the discovery that the LRRK2 gene causes PD, it has become clear that this protein is a prominent molecular target for disease-modifying efforts. The clinical development of small molecule LRRK2 kinase inhibitors is an important step forward, making this an exciting time for the LRRK2 field. However, in parallel with the development of these drugs, we still need to understand the as-yet-undetermined neuropathological basis of LRRK2-PD. It is equally important to develop a nuanced understanding of the LRRK2-mediated early synaptic events that precede the disease symptomatology. A crucial emerging concept is that the cellular phenotypes caused by the disease-linked mutations in LRRK2 vary across brain regions and cell types. For example, LRRK2 impairs synaptic vesicle trafficking in dopaminergic, but not glutamatergic, neurons [77]. Meanwhile, the alterations to excitatory postsynaptic currents observed in the dorsomedial striatum are not identical to those observed in the ventral striatum [76,79]. Within the dorsal striatum, the postsynaptic structures of direct- and indirect-pathway neurons are also differentially affected [78]. These observations dovetail with prior observations of cell-type-specific changes in PD, such as the changes to morphology and excitability that occur in indirect- and direct-pathway SPNs in response to dopamine loss [132]. 

Some outstanding questions about these synaptic changes include: (1) The mechanisms of cell-type specificity, i.e., why are changes to neurotransmitter release and glutamatergic postsynaptic densities observed in some cell types and not others? (2) Relatedly, what is the extent of the cell-type specificity? For example, there are clear differences between LRRK2-associated phenotypes in glutamatergic and dopaminergic neurons, but recent work has identified multiple subtypes of dopaminergic projection neurons within the SNc. Future work should aim to resolve whether PD risk genes including LRRK2 differentially affect specific dopamine neuron subtypes, and how LRRK2 function in dopaminergic neuron differs from other neuromodulatory projection neurons. (3) To what extent are these changes reversible? Do they represent the effects of LRRK2 during a critical developmental window, or the ongoing effects of LRRK2 in cell biology? Finally, and most importantly, (4) do these early synaptic changes occur independently from the development of PD, or do they contribute to the development of pathology later in life? If so, do they represent an opportunity for early therapeutic intervention? 

We have focused primarily on cell-type specificity within striatal neurons. However, the differential vulnerability of midbrain dopaminergic neurons in PD, combined with what we now know about the cell-type-specific effects of LRRK2 mutation in other regions and cell types, compels a finer dissection of the differential effects of LRRK2 among neuron subtypes in the dopaminergic midbrain. Genetic models that offer the opportunity to perform cell-type-specific manipulations of LRRK2 in dopaminergic neurons, including Cre and inducible driver lines as well as mouse lines allowing the Cre-dependent expression of LRRK2 mutant alleles, already exist. In addition to the areas highlighted in this review, these models can be applied to studies of the cellular changes occurring later in life that have been more explicitly linked to neurodegeneration to answer questions about the specific vulnerability of dopaminergic neurons in PD. 

## Figures and Tables

**Figure 2 cells-11-00169-f002:**
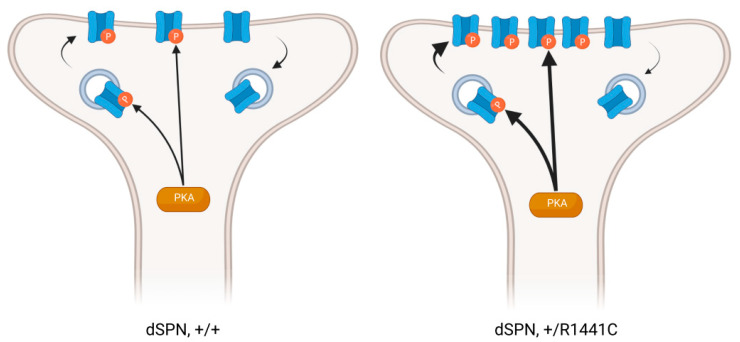
Pathway-specific effect of LRRK2 on synaptic AMPARs. The R1441C mutation induces an enhanced phosphorylation of GluR1 by PKA, leading to the increased insertion and retention of AMPA receptors into the synaptic membrane of dSPNs, but not iSPNs [78]. This figure was created using biorender.com.

**Figure 3 cells-11-00169-f003:**
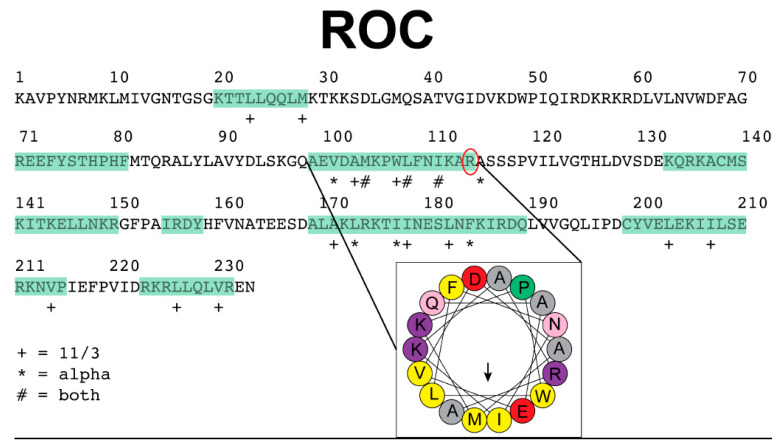
Candidate AKAP-like helix in the LRRK2 ROC domain. The sequence of the ROC domain is presented with alpha helices identified from sequence analysis highlighted in green. A candidate amphipathic helix was identified within the ROC domain of LRRK2 and is highlighted by the black lines leading to the inset image of the helical wheel. This helix contains the clinically relevant residue R1441 (circled in red). In the legend the “+” indicates resides belonging to a hydrophobic face when a 3–11 helix is generated. The “*” indicates residues belonging to the hydrophobic face when an alpha helix is modeled. The “#” identifies residues that belong to the hydrophobic face when either a 3–11 or alpha helix is generated.

**Figure 4 cells-11-00169-f004:**
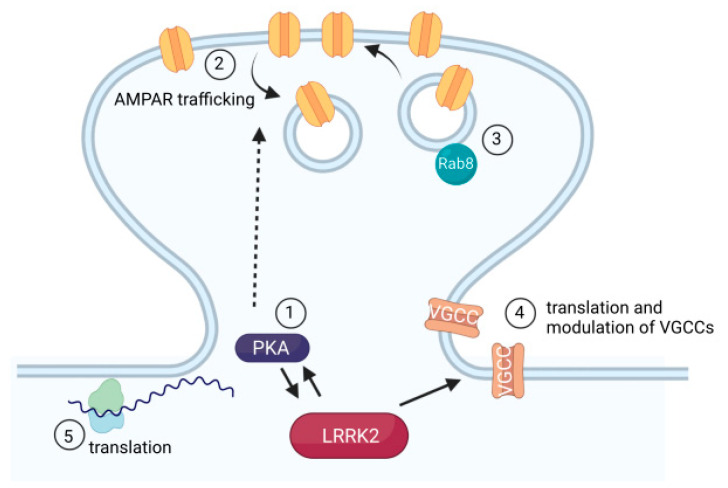
Effectors of LRRK2 postsynaptic function. Depiction of mechanisms by which LRRK2 affects synaptic function and downstream LRRK2 effectors that affect synapse function. (1) An interaction between the PKA PKARIIβ subunit and LRRK2 ROC domain acts as an AKAP, anchoring LRRK2 in dendritic shafts. The R1441C mutation impairs this interaction, resulting in increased translocation of PKA into dendritic spines. (2) Mutation- and cell-type-specific alteration of AMPAR trafficking in dendritic spines. The R1441C mutation increases GluR1 phosphorylation and AMPAR insertion into the membrane specifically in direct-pathway SPNs, increasing synapse strength. (3) The LRRK2 substrate Rab8 is essential for trafficking newly translated AMPARs to the synapse. (4) LRRK2 kinase activity modulates the translation (by affecting the efficiency with which ribosomes translate mRNAs with complex 5′ UTRs), and the function of voltage-gated calcium channels (VGCCs), resulting in increased calcium influx. (5) LRRK2 phosphorylates the ribosomal S15 subunit, promoting the increased translation of mRNAs. This figure was created using biorender.com.

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
