# Peer review of "LRRK2 at Striatal Synapses: Cell-Type Specificity and Mechanistic Insights"

_cells, 2022, doi:10.3390/cells11010169_

Round 1

Reviewer 1 Report

The review entitled “LRRK2 at striatal synapses: Cell-type specificity and mechanistic insights” by Patrick Skelton et al., provides a comprehensive coverage of the synaptic aberrations caused by LRRK2-related mutations. Particularly, both pre- and post-synaptic mechanisms that affect synaptic functions. Attempt had also been made to link mechanistic with single cell RNA-seq work to pinpoint cause of neurodegeneration at single cell level. This is considered novel.

However, the overall flow for each section is not clear. There appears to be overlapping or confounding information appearing periodically between certain sections. Author should consider rearranging similar information and aligning them with an appropriate subheading and objective within each section to improve the overall flow of the paper. Furthermore, detailed information is lacking on single cell cell-type specificity in contributing to synapses dysfunction in LRRK2-linked PD.

My specific comments are as followings:

  1. Suggest to include introduction section to provide reader an overview
  2. Structure of the paper
  • Suggest to include more subheadings especially for [Line: 198-348]. Suggest the use of sub-headings to further segregate information.
  • Suggest to change heading in [Line: 190] to “LRRK2’s mutants modulate synapse specific function” so as to in line with the content
  • Suggest the inclusion of sub-headings in subsequent paragraph. For example, “delineate information between paragraphs based on synapses classified by their neuromodulators”
  • In the section “Cell-type specificity of LRRK2 expression” [Line:112], it can be further segregated based on an appropriate overarching theme. (ie: Neuromodulator classification; Dopaminergic, Cholinergic, GABAergic)
  1. Section 3 – Cell-type specificity of LRRK2 expression (Line 112 – 189).
  • Data from single-cell expression only mentions specificity of LRRK2 expression across different cellular populations of the CNS with spatial specificity with respect to cortical areas. Suggest to include snippets of temporal information, since it appears that LRRK2 expression in young mice is also critical towards disease development.
  • In addition, there are brief mentions of the importance of differing temporal window affecting results interpretation but no specific information is included (Post-natal (P) days). Suggest to elaborate with more detailed information, since they are important and have already cited the relevant references
  • Suggest to elaborate cell type specificity and signaling pathways link to mechanistic insights underlying PD.

  1. Section 4 – Evidence for synaptic functions (Line 190 – 348), more information required:
  • [Line 227] Any evidence for animal models with transgenic LRRK2 comparing dysregulated electrophysiology in young mice, which then proceed to show PD-like pathology in later periods?
  • [Line 244-246] Are there any cortical areas that is disproportionately affected with respect to the altered electrophysiology. Furthermore, is there any evidence to point towards a feedback mechanism or network architecture between in cortical projection neurons that could support the phenomenon observed?
  • [Line 346-347] Authors rightly pointed out that more in-depth systematic approaches are required in deriving more conclusive conclusions. Perhaps, the authors can provide some examples on how these approaches can be implemented and how their suggested method is able to provide a better understanding of the current knowledge gap as identified.
  1. Section 5 – LRRK2 Postsynaptic mechanisms (Line 349 – 543), more clarification required:
  • [Line 383 – 418] The portion on the ROC domain could be streamlined to delve straight into the point that the author is trying to bring across.
  • Furthermore, Fig.3 appears to be a little out of place, other than showing the associated sequence responsible for the amphipathic helix. Suggest to stay relevant to the theme of the section and not deviate or delve into unnecessary information that may not be pertinent to the development of the story. Suggest to summarize and phrase in a more concise manner. (ie: Main point is to reference that LRRK2::AKAP is necessary for PKA interaction, and that this AKAP domain functions through an amphipathic helix). Next, include how the 2 prominent mutations of LRRK2 affects this binding interaction through alterations to the amphipathic helix. Other information that is not pertinent to the development of this punchline could be trimmed.
  • Also, the authors can include more direct relation to how this interaction is related to post-synaptic functions by further strengthening the point of involvement in interaction between LRRK2 and PKA.

Minor:

  1. Figure 1.
  • Axis-legend for bottom right panel in Figure 1(A) is missing for some subclusters identified iSPNs and dSPNs.
  • Formatting of figure legend. Include links to related datasets on dropviz as brackets for Panel 1(A) and 1(B), respectively.
  1. Figures legends. Suggest to include more detailed description or illustrations that were consistent with relevant information described in the paragraphs text.
  2. Adjustments between figure captions and naming convections of certain genotypes could be formatted and standardized throughout the paper. If possible, do include the fully defined terminology for atypical acronyms.

Reviewer 2 Report

This is a generally well written review that will be of interest to those studying LRRK2 mutations and their consequences in the nigrostriatal circuit.  Overall the discussion is balanced although Pages 1-3 are highly redundant with many other similar reviews.  I have only minor suggestions.

  1. Line 7.  Mutations in LRRK2 (not to)
  2. Line 86 makes no sense
  3. Line 181 should refer to ref. 55,56 not 57; Line 187 is ref. 55,56
  4. Line 360 needs a reference
  5. Line 427 is controversial in terms of which kinase phosphorylates S935--the authors should cite more papers here
  6. Line 458 Golgi should be capitalized
  7. The authors conflate dominant negative Rab consequences with phosphoRabs.  This is probably not correct as only a small proportion of total Rabs are phosphorylated in any cell.
  8. The review does not mention PPM1H phosphatase, the abundance and cell type expression of which is just as important as the level of LRRK2. 
  9. Otherwise, well done!

Round 2

Reviewer 1 Report

In the revised version, the authors have adequately addressed concerns pertaining to the first review. Minor changes pertaining to formatting and proofreading is required, otherwise there is no major issues with the paper in its current format. No other comments as indicated.